# Modelling the effectiveness of antiviral treatment strategies to prevent household transmission of acute respiratory viruses

Hind Zaaraoui[1]*, Clarisse Schumer[1], Xavier Duval[1], Bruno Hoen[2], Lulla Opatowski[3,4]ʘ, Jérémie Guedj[1]ʘ

1 Université Paris Cité, IAME, INSERM, F-75018 Paris, France, 2 Campus Brabois Santé, Université de Lorraine, Nancy, France, 3 Epidemiology and Modelling of Antimicrobials Evasion research unit, Institut Pasteur, Paris, France, 4 Université Paris-Saclay, UVSQ, INSERM, CESP, Anti-infective evasion and pharmacoepidemiology research team, Montigny-Le-Bretonneux, France

ʘ These authors contributed equally to this work.
* hind.zaaraoui@inserm.fr

**Data Availability Statement:** For the viral load dynamic model parameters' calibration, we used public available data from https://github.com/

## Abstract

Households are a major driver of transmission of acute respiratory viruses, such as SARS-CoV-2 or Influenza. Until now antiviral treatments have mostly been used as a curative treatment in symptomatic individuals. During an outbreak, more aggressive strategies involving pre- or post-exposure prophylaxis (PrEP or PEP) could be employed to further reduce the risk of severe disease but also prevent transmission to household contacts. In order to understand the effectiveness of such strategies and the factors that may modulate them, we developed a multi-scale model that follows the infection at both the individual-level (viral dynamics) and the population-level (transmission dynamics) in households. Using a simulation study we explored different antiviral treatment strategies, evaluating their effectiveness on reducing the transmission risk and the virological burden in households for a range of virus characteristics (e.g., secondary attack rate—SAR, or time to peak viral load). We found that when the index case can be identified and treated before symptom onset, both transmission and virological burden are reduced by > 75% for most SAR values and time to peak viral load, with minimal benefit to treat additionally household contacts. While treatment initiated after index symptom onset does not reduce the risk of transmission, it can still reduce the virological burden in the household, a proxy for severe disease and subsequent transmission risk outside the household. In that case optimal strategies involve treatment of both index case and household contacts as PEP, with efficacy > 50% when peak viral load occurs after symptom onset, and 30-50% otherwise. In all the considered cases, antiviral treatment strategies were optimal for SAR ranging 20-60%, and for larger household sizes. This study highlights the opportunity of antiviral drug-based interventions in households during an outbreak to minimize viral transmission and disease burden.

lacyk3/SARS-CoV-2Kinetics. All the results' computation are simulated thanks to the Matlab code submitted in the S1 Code (see code and Readme file). The code is protected by Agence de Protection des Programmes (APP) in France, and we have the permission to share it.

**Funding:** This project received funding from the European Union's Horizon 2020 research and innovation program under grant agreement No 101016167, ORCHESTRA (Connecting European Cohorts to Increase Common and Effective Response to SARS-CoV-2 Pandemic). The funders had no role in study design, data collection and analysis, decision to publish, or preparation of the manuscript.

**Competing interests:** The authors have declared that no competing interests exist.

## Author summary

### Why was this study done?

- The attack rate of acute respiratory viruses is particularly high in households. Antiviral treatments can be strategically used to break transmission chains and reduce virus burden, however their use is limited by the difficulty to understand the factors that determine their effectiveness.

- Two distinct antiviral strategies, either treating the index individual upon diagnosis, or treating all household members upon index diagnosis, can be evaluated in terms of their reduction of viral infections and virological burden in the household.

### What did we do and find?

- We developed a theoretical framework to determine the impact of antiviral treatment in households, integrating the complex interplay between viral load dynamics, antiviral treatment initiation with respect to symptom onset and peak viral load, secondary attack rate and household size.

- All strategies reduce the number of infections by more than 50% when antiviral treatment can be be administered before symptom onset of the index case, but not after.

- Treatment initiated after symptom onset are effective in reducing the virological burden in household, in particular when both index cases and household contacts are treated and peak viral load occurs after symptom onset.

- The effectiveness of antiviral treatment strategies is larger for SAR ranging between 20 and 80%, and increases with the size of the household.

### What do these findings mean?

- Our work offers a theoretical framework to anticipate the effectiveness of antiviral strategies in households. It can be used to optimize prescription strategies aiming to prevent viral transmission and reduce disease burden in households.

## Introduction

Households are the epicenter of community transmission of acute respiratory viruses, such as Influenza or SARS-CoV-2, with transmission rates resulting from close, repeated and inter-generational interactions [1, 3, 45]. For instance, in the case of pre-Omicron SARS-CoV-2 virus, the household attack rate, defined as the fraction of infected individuals after infection of an index case, is close to 40%, albeit with large variations across studies reflecting the hetero-geneity in pre-existing immunity, social habits, as well as size and composition of households [4, 6, 7, 19]. In the Omicron variant era, values as large as 80% have been reported, reflecting

the dominance of even more transmissible viruses that can escape immunity conferred by vaccine or previous infection [7]. Finding ways to prevent the transmission of respiratory viruses in households is therefore key to prevent community transmission and, in case of threat to public health, avoid to resort to interventions with an unsustainable economic and social, such as lockdown, curfew or school closure.

Given the versatility of respiratory viruses, preventing community transmission of severe infections cannot rely on a single, albeit effective, intervention but rather requires a combination of non-pharmacological and pharmacological interventions that, together, can reduce the risk of severe disease in individuals with high-risk factors and break the chains of transmissions. One important but still under-employed layer of defense is the use of antiviral treatments, such as monoclonal antibodies or small molecules. As exemplified for SARS-CoV-2, these treatments can be highly effective in treating infected patients, reducing the risk of severe disease by 70–90% when administered within the first week of symptom onset [8, 9, 16], but they can also be used as pre- or post-exposure prophylaxis to reduce the risk of symptomatic disease [10, 11] or transmission. This approach is not novel, and large-scale treatment strategies in households have already been proposed in the past to reduce the burden of seasonal or pandemic Influenza virus [13, 14]. In the context of SARS-CoV-2, such strategies have been so far limited by a limited supply and the need for many of these treatments to be administered intravenously but the rapid development of safe, effective and orally available drugs, such as Paxlovid [2, 12], could broaden their use.

The large deployment of antiviral treatment strategies remains however limited by the difficulty to measure their effectiveness and to factor in the multiple parameters that modulate it, in particular the virus transmission rate and the timing of treatment administration. While "the sooner, the better" is a long-standing paradigm of antiviral therapy [15], its implementation in real life is hampered by the challenge to identify patients before large quantities of virus have already been excreted. In the future, the experience acquired during the last pandemic in terms of contact tracing and implementation of contactless clinical studies [42] will enable more precocious strategies that can be implemented in case of threat to public health. This, therefore, makes it urgent to clearly identify the conditions required for a successive deployment of antivirals in households, and to quantify the effectiveness that can be expected.

We here define a quantitative framework to predict the effectiveness of antiviral treatment strategies in households. We develop a multi-scale model integrating both the evolution of viral load within infected individuals and the risk of virus transmission. By factoring in the relative role of the epidemiological, clinical, pharmacological, virological and immunological parameters and by using the model to simulate thousands of households, we evaluate the impact of antiviral treatment under different prescription scenarios/strategies to mitigate acute respiratory virus transmission in households.

## Methods

### A multi-scale model to link viral dynamics and the risk of transmission over time

**Viral dynamic model.**    The viral dynamic (within-host) model characterizes the change in viral load levels after infection at time $t = 0$. It builds on previous model developed for SARS-CoV-2 and other acute viral infections [20, 21, 34]. In brief, the model includes three types of cell populations: uninfected susceptible target cells ($T$), infected cells in an eclipse phase ($I_1$), and productively infected cells ($I_2$). The model assumes that target cells are infected at a constant rate $\beta$. Once infected, cells enter an eclipse phase and become productively infected at a constant rate $k$. Productively infected cells produce viral particles at a constant rate $\pi$ and are

eliminated at a dynamic rate $\Delta(t)$ where $\Delta(t) = \delta_1$ before adaptive response time $\tau$, and $\Delta(t) = \delta_2$ when $t \geq \tau$. A fraction $\mu$ of the viral particles is infectious, noted $V_I$, and the remaining viral particles are non infectious, noted $V_{NI}$. Viral load at time $t$ post infection, $V(t)$, is the sum of infectious and non-infectious viral particles, both cleared at the same rate constant $c$. In addition, the model accounts for a time-dependent immune response via a dimensionless compartment, $F$, which is stimulated by the presence of viral particles. In this model, $F$ has an intrinsic loss rate, noted $d_F$, and $F$ acts by increasing the loss of infected cells, with a non-linear and saturable effect defined by $\frac{F}{F+\theta}$ where $\theta$ is the level of $F$ required to achieve 50% of the maximum immune response. The model also incorporates an adaptive immune response with a refractory state. The innate immune response is modelled through the refractory compartment $R$ and the effect of IFNs. The transition rate of cells from the susceptible state to refractory one is defined by the parameter $\phi$. Cells within the refractory compartment can come back with rate $\rho$ to the susceptible target compartment, providing temporary protection induced by IFNs. The model is given by the following equations:

$$\frac{dT}{dt} = -\beta TV - \phi TF + \rho R$$

$$\frac{dR}{dt} = \phi FT - \rho R$$

$$\frac{dI_1}{dt} = \beta VT - kI_1$$

$$\frac{dI_2}{dt} = kI_1 - \Delta(t)I_2$$

$$\frac{dV_I}{dt} = \pi\mu\left(1 - \frac{F}{F+\theta}\right)I_2 - cV_I \tag{1}$$

$$\frac{dV_{NI}}{dt} = \pi(1 - \mu)\left(1 - \frac{F}{F+\theta}\right)I_2 - cV_{NI}$$

$$\frac{dF}{dt} = qI_2 - d_f F$$

$$\text{where} \begin{cases} \Delta(t) = \delta_1 & t < \tau \\ \Delta(t) = \delta_2 & t \geq \tau. \end{cases}$$

The basic within-host reproduction number, $R_0^{intra}$, defined as the average number of secondary infected cells resulting from one infected cell in a population of fully susceptible target cells, is equal to $\frac{pT_0\beta\mu}{c\delta}$.

**Transmission model.** Following previous publications [24], a Power-law model is used to relate the non-linear relationship between viral load at time $t$, to the instantaneous risk of transmission during a high-risk contact, $p(t)$:

$$p(t) = 1 - e^{-M \times V(t)^h} \tag{2}$$

where $M$ quantifies the strength of the association between viral load and transmission and $h$ reflects the stiffness of this association. Another way at looking at the model is to observe that $p(t) = 1 - \left(e^{-V(t)^h}\right)^M$, such that $M$ can also be interpreted as a proxy of the intensity of the contact. To account for the variability in this parameter across individuals, due to different behavioral or biological factors, we further assumed that $M$ follows a log-normal distribution with mean value $m$ and standard deviation $\sigma$. As household contacts are not unique and may be repeated, we note $P(\infty) = \lim_{t\to\infty} P(t)$ the probability that at least on these contacts leads to an

infection, given by: $P(t) = 1 - \prod_{u \in [0,t]}(1 - p(u)))$, and we assume without loss of generality 1 contact every 12 hours (i.e., 2 contacts per day). Finally we define the Secondary Attack Rate (SAR) as the mean of $P(\infty)$ over all infected individuals.

**Impact of antiviral treatment on viral load.** When an individual receives an antiviral treatment at time $t = t_x$ after infection, the viral dynamic model given by the equations differential system (1) is modified to reflect the effect of treatment on reducing the production of viruses by infected cells, with an efficacy noted $\epsilon$, leading to the following model:

$$\frac{dT}{dt} = -\beta TV - \phi TF + \rho R$$

$$\frac{dR}{dt} = \phi FT - \rho R$$

$$\frac{dI_1}{dt} = \beta VT - kI_1$$

$$\frac{dI_2}{dt} = kI_1 - \Delta(t)I_2$$

$$\frac{dV_I}{dt} = \pi(1 - \epsilon)\mu\left(1 - \frac{F}{F + \theta}\right)I_2 - cV_I \tag{3}$$

$$\frac{dV_{NI}}{dt} = \pi(1 - \epsilon)(1 - \mu)\left(1 - \frac{F}{F + \theta}\right)I_2 - cV_{NI}$$

$$\frac{dF}{dt} = qI_2 - d_f F$$

$$\text{where } \begin{cases} \Delta(t) = \delta_1 & t < \tau \\ \Delta(t) = \delta_2 & t \geq \tau. \end{cases}$$

Importantly, we here assume that treatment, once initiated, is continued until virus eradication. By reducing viral replication, treatment reduces viral load levels and hence the risk of transmission as given by Eq (2) (see Figs 1 and 2).

**Model calibration.** To calibrate our viral dynamic model, we used data from the National Basketball Association's cohort [17, 18], which constitutes the most detailed dataset available on SARS-CoV-2 viral load so far. Overall the model was fitted to 607 individuals, that could be infected with pre-Omicron or Omicron variants. Inference procedure is detailed in S1 Text. *"Viral load dynamic model and calibration"* and parameter values are summarized in S1 Table.

In addition, we assume that the incubation period follows a log-normal distribution with a mean of 4 days and a standard deviation of 0.125 days, i.e., 90% of patients have an incubation period ranging between 3 and 6 days as observed for SARS-CoV-2 [26]. We assume that treatment antiviral efficacy, $\epsilon$, is equal to 99% on average (S1 Table), as observed for highly potent protease inhibitors [12].

Regarding the parameters governing the relationship between the viral load and the risk of transmission in Eq (2), we assume homogeneous mixing in the household and that all individuals have two (high risk) contacts per day to any other household member. We fix the value of $h$ to 0.49, as previously estimated [24]. The parameter $M$ is sampled from a log-normal distribution with mean $m$ and standard deviation $\sigma = 0.85$ (see S1 Table). We consider $m$ values ranging from $2.9 \times 10^{-7}$ to $7.9 \times 10^{-5}$ mL/cp in order to reproduce SAR ranging from 5% to 97% in absence of antiviral treatment (Fig 2D).

The joint within- and between- host model can generate usual metrics of epidemiological studies, such as the SAR or the generation time, defined as the interval between the infection of an index infector and the infection time of its secondary cases. For instance, $m = 3.76 \times 10^{-6}$

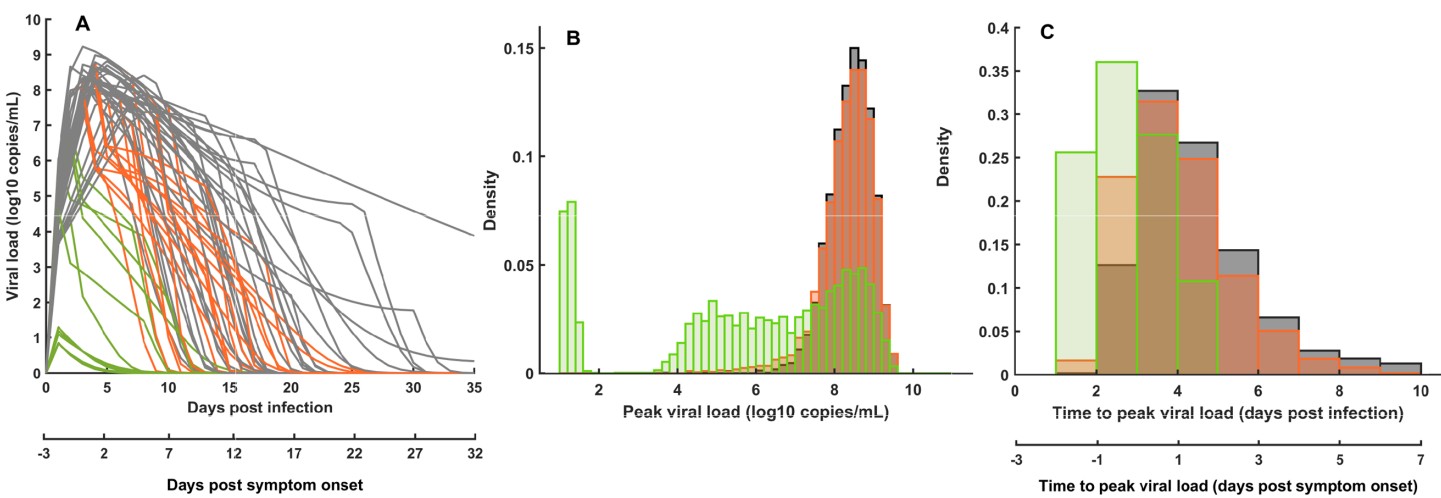

**Fig 1. Viral dynamics and antiviral treatment.** (**A**) Individual viral dynamic profiles predicted by the model in 30 individuals (Eq 3), that are either untreated ($I_0$, gray), treated within 5 days after symptom onset ($I_{cur}$, orange), or treated within 4 days after infection ($I_{pep}$, green). All parameters are given in S1 Table, and the model assumes that a mean incubation period of 4 days, and a mean treatment antiviral efficacy of 99%. (**B**) Distribution of the peak viral load predicted by the model. (**C**) Distribution of the time to peak viral load predicted by the model. Note that gray and orange distributions overlap in (**B**) and (**C**) as the treatment is mostly initiated after the peak load time.

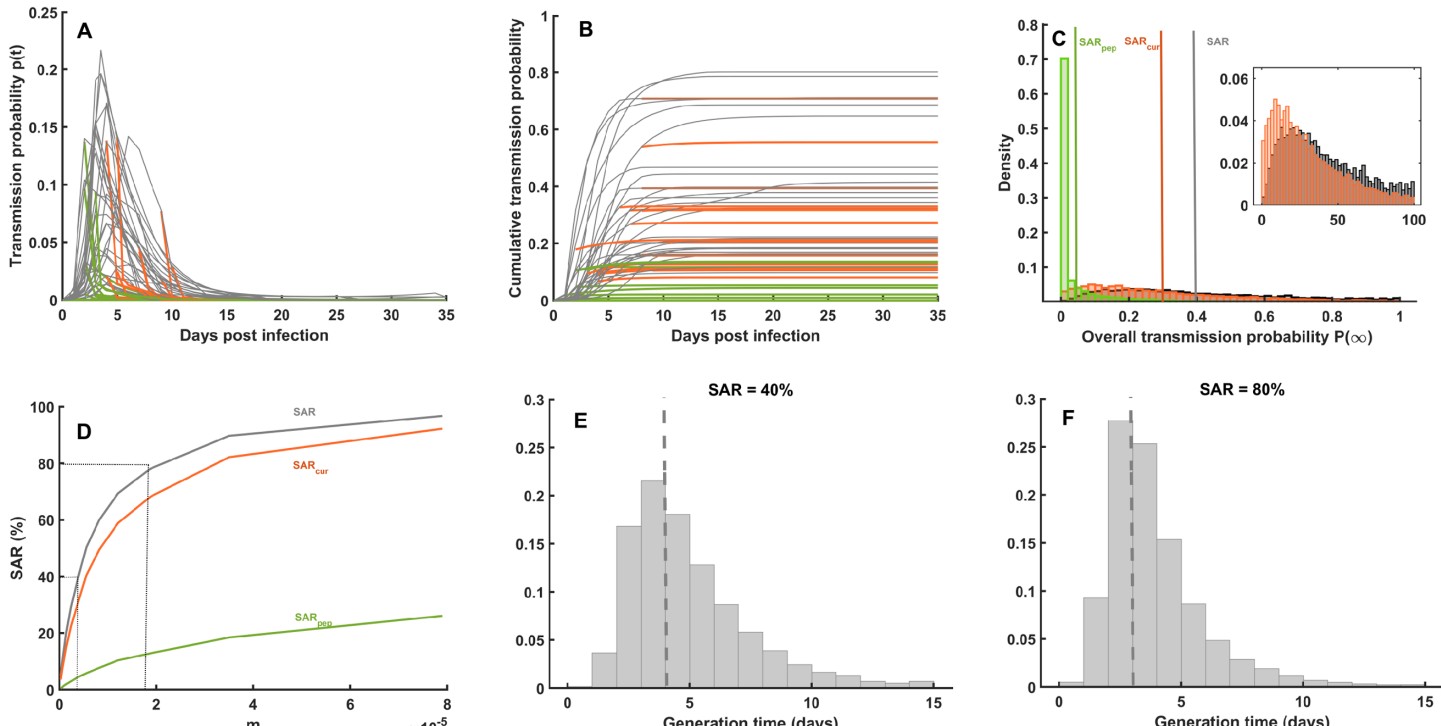

**Fig 2. Impact of antiviral treatment on the risk of virus transmission.** (**A**) Instantaneous probability of virus transmission, $p(t)$, in 30 individuals that are either untreated ($I_0$, gray), treated within 5 days from symptom onset ($I_{cur}$, orange), or treated within 4 days after infection ($I_{pep}$, green). All parameters are given in S1 Table. (**B**) Cumulative probability of transmission to another member of the household, $P(t)$, assuming two contacts per day. (**C**) Distribution of the overall transmission probability, $P(\infty)$. Top right, without treatment, with SAR defined as the mean of the distribution. Similar definitions are used to defined SAR when treatment is initiated after symptom onset ($SAR_{cur}$, orange) or before symptom onset ($SAR_{pep}$, green). (**D**) SAR, $SAR_{cur}$, and $SAR_{pep}$ values according to different values of $m$. (**E, F**) Generation time distribution for SAR = 40% and SAR = 80% respectively. **A-B**: Simulations conducted for $m = 3.76 \times 10^6$ mL/copies (see Eq (2)), corresponding to SAR = 40%.

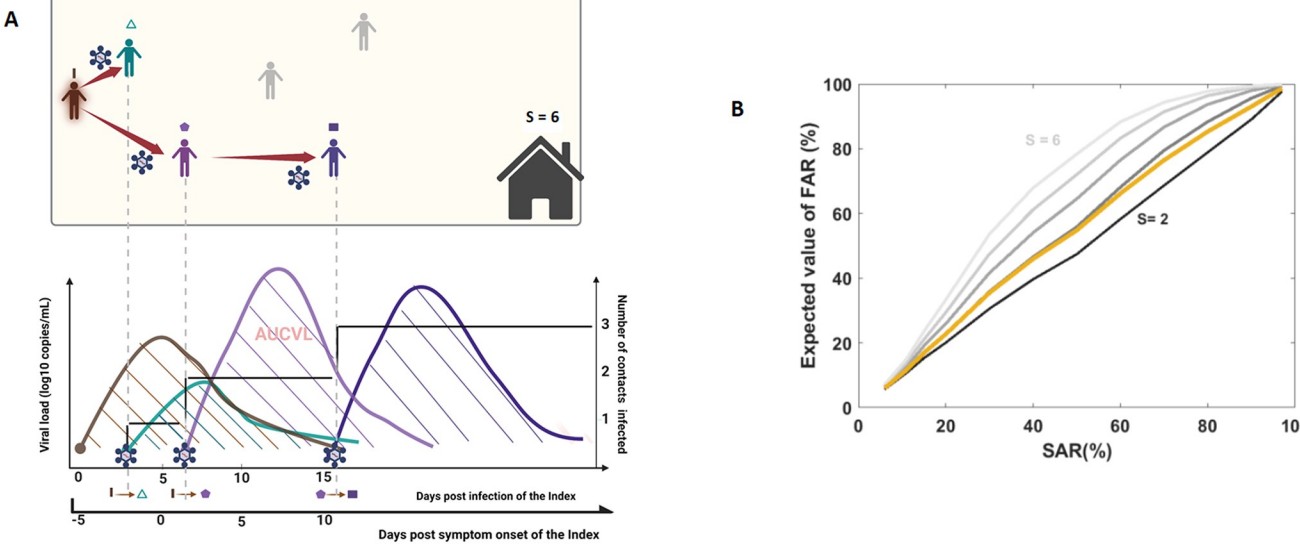

**Fig 3. Transmission chains in households.** (**A**) Top: Illustrative schematic of a transmission chain in a household; Bottom: temporal profile of the virological burden. The dashed area represents the cumulative area under the curve of the viral load in the household (**these two figures were created with BioRender.com**). (**B**) Model based prediction of the relationship between the Final Attack rate (FAR) and SAR, for household sizes ranging from 2 to 6. The yellow line represents the average FAR when sampling in the household size distribution in France.

mL/copies corresponds to SAR = 40%, as typically observed with SARS-CoV-2 pre-Omicron variants [31], leading to a mean generation interval of 4 days, consistent with values reported in the literature [47–50]. Similar calculations can be done to reproduce values observed with SARS-CoV-2 Omicron variants (Fig 2).

## Modeling household transmission and impact of antiviral treatment

**Household transmission and measure of the outbreak severity.** The within- and between-host model presented until now only considers the risk of transmission from one infected individual to another individual, but it does not account for transmission chains that can occur in households of size $S > 2$ (Fig 3A). We therefore generalize the previous model to households of size $S > 2$ assuming homogeneous mixing and fixed contact rate between all household members. Therefore, transmission can occur to any other non-infected individuals in the household, and we assume that an individual can be infected only once during an outbreak (no reinfection). When all transmission chains in the household have extinguished, outbreak severity in a household $h$ can be measured using two metrics (Fig 3A):

- The number of cases, noted $c_h$, from which one can define the transmission rate, defined as the proportion of secondary infections in the household, also called the Final Attack Rate (FAR), as $FAR_h = \frac{c_h - 1}{S - 1}$.

- The total virological burden, $AUCVL_h$, computed as the sum of the area under the curve of the viral load of all infected individuals in the household. Because $AUCVL_h$ measures the total amount virus that has been excreted by a household during an outbreak, it can be used as a proxy of the risk of both severe infection (within the household) and of virus transmission in the community (outside the household) as, $AUCVL_h = \sum_{k=1}^{c_h} \int_0^\infty V_{k,h}(u)du$ where $V_{k,h}(u)$ the viral load at time $u$ for individual $k$ in the household $h$.

**Intervention strategies and measure of their effectiveness.** We consider four treatment strategies. In the first scenario, noted $I_{cur}$, the index individual initiates treatment within 5 days from symptom onset, assuming a uniform distribution for treatment initiation between 0 and 5 days post symptom onset. The second scenario, noted $I_{pep}$, assumes that treatment is administered before symptom onset as a post-exposure prophylaxis, assuming again a uniform distribution for treatment initiation between infection time (t = 0) and symptom onset. Then we consider the same treatment strategies, assuming now that treatment is given not only to the index individual, but also and simultaneously to all household members, regardless of their infection status, as either a post-exposure prophylaxis (when the index is already symptomatic, noted $I_{cur} + H_{pep}$), or as a pre-exposure prophylaxis (when the index is not yet symptomatic, noted $I_{pep} + H_{prep}$) (Table 1).

Next, for each of these strategies, we simulated outbreaks in a large number of households, $N_{sim}$, using the distribution of household sizes in France (see below) and this provided us with an (expected) number of infection ($N_{inf} = \frac{1}{N_{sim}} \times \sum_{h=1}^{N_{sim}} c_h$), virological burden ($AUCVL = \frac{1}{N_{sim}} \times \sum_{h=1}^{N_{sim}} AUCVL_h$) and transmission rate: $FAR = \frac{1}{N_{sim}} \times \sum_{h=1}^{N_{sim}} FAR_h$.

We then calculated the effectiveness of each strategy $I_{tx}$, as the relative reduction in the mean outcome compared to that obtained when no treatment is given, $I_0$. Thus, an effectiveness of 50% in the transmission rate implies that a strategy reduces by half the mean FAR as compared to a strategy where no treatment is used ($1 - \frac{FAR^{I_{tx}}}{FAR^{I_0}}$).

**Table 1. Main assumptions of the model.**

| Model component | Main model | Alternative models |
|---|---|---|
| **Viral dynamic model** | | |
| Viral kinetic model | Target cell model with an innate immune response | NA |
| Mean time to peak viral load | 4 days | 1 and 7 days |
| Mean time to symptom onset (Incubation) | 4 days | NA |
| **Transmission model** | | |
| Probability of virus transmission during a high risk contact at time $t$ | $p(t) = 1 - e^{M \times V(t)^{0.49}}$ <br> $M \sim \log\text{-}\mathcal{N}(m, 0.85^2)$ | NA |
| Secondary Attack Rate (SAR) | 40% ($m = 3.76 \times 10^{-6}$ mL/cp) <br> 80% ($m = 1.88 \times 10^{-5}$ mL/cp) | 5–97% <br> ($m \in [2.9 \times 10^{-7}, 7.9 \times 10^{-5}]$ mL/cp) |
| Household size (S) | 2 to 6 <br> (distribution based on French demographics) | NA |
| Number of high risk contacts between any two household members | 2 per day | NA |
| Impact of household size on SAR | none | SAR decreases with S |
| **Antiviral treatment** | | |
| Mechanism of action | Inhibits virus production | NA |
| Antiviral efficacy | 99% | 50% and 90% |
| **Treatment strategies** | **Index individual** | **Household members** |
| $I_0$ (reference) | No treatment | No treatment |
| $I_{cur}$ | Treatment randomly initiated within 5 days from symptom onset | No treatment |
| $I_{pep}$ | Treatment randomly initiated before symptom onset | No treatment |
| $I_{cur} + H_{pep}$ | Treatment randomly initiated within 5 days from symptom onset | Simultaneous initiation in all household members |
| $I_{pep} + H_{prep}$ | Treatment randomly initiated before symptom onset | Simultaneous initiation in all household members |

We also calculated for each strategy $I_{tx}$ the number of cases avoided per number of treatments administered as the reduction in the total number of infected individuals as compared to no treatment ($I_0$), divided by the total number treatment courses given, noted $N_{tot} = \sum_{h=1}^{N_{sim}} n_h$, as

$$\frac{1}{N_{tot}} \times \sum_{h=1}^{N_{sim}} (c_h^{I_0} - c_h^{I_{tx}}) \tag{4}$$

where $n_h$ is the number of treatment given in household $h$ which can be either equal to 1 ($I_{cur}$ or $I_{pep}$ strategies) or $S_h$, the size of the household h ($I_{cur} + H_{pep}$ or $I_{pep} + H_{prep}$ strategies)

**Household distribution.** We simulated the outbreak in $N_{sim}$ = 50, 000 households for each antiviral strategy considered. Because results depend on household size, we rely on the distribution of household sizes reported by the French National Institute of Statistics and Economic Studies (INSEE) [33] with households of size 2, 3, 4, 5 and 6 representing 50.85%, 21.62%, 18.20%, 6.69% and 2.64% of all households, respectively (see S2 Table).

**Designing a clinical trial to evaluate the effectiveness of an antiviral strategy.** We finally evaluate the number of households that would be required to demonstrate in a clinical trial a significant effectiveness of the different antiviral strategies, considering as primary endpoints either the mean Final Attack Rate (FAR) or the mean virological burden (AUCVL). We consider a controlled study versus placebo (no treatment), with a 1:1 randomization on households, a type I error of 5% (two-sided test) and a statistical power of 90% for. AUCVL was treated as a continuous random variable normally distributed and a t-test was used to assess difference between treated and untreated, while FAR was analyzed as a discrete random variable using a binomial test.

## Generalization of the model and sensitivity analysis

**Evaluating the impact of different viral kinetic patterns and viruses.** Because the effectiveness of each strategy critically depends on the delay between treatment initiation and peak viral load, we finally generalize our approach to different patterns of viral kinetics. In addition to the initial scenario where the time to peak viral load coincides with symptom onset and occurs at 4 days post infection, we now also consider more general cases where the time to peak viral load occurs at 1 or 7 days post infection, i.e., 3 days before or after symptom onset, respectively. Parameter values used to reproduce these kinetics are summarized in S3 Table.

**Exploring a secondary Attack Rate that depends on household size.** Importantly, our model assumes that SAR is independent of the household size. As a large household size increases the risk of tertiary transmission (i.e., not originating from the index case), our model predicts that FAR increases with household size (Fig 3B). This may however overlook the existence of mechanisms that limit the spread of the virus in large households, such as the fact that contact rate are not homogeneous, as observed in many epidemiological studies [19, 27–30, 32].

We therefore also considered in a sensitivity analysis a model where SAR decreases with the size of the household size. Noting SAR the value observed for $S$ = 2, as before, we note $SAR_S$ the SAR for household size greater than 2 and we assume that $SAR_S = \frac{SAR}{\sqrt{S-1}}$ ($S \geq 3$). Thus, in this setting, we used a different value of m in each household size, relying on the relationship between $m$ and SAR that was established above. Simulations show that the expected value of FAR is much less sensitive to the household sizes (see S1 Fig).

**Drug antiviral efficacy.** In our main analysis we assume that treatment antiviral efficacy, $\epsilon$, is equal to 99% on average (S1 Table) [12]. As a sensitivity analysis, we also consider lower antiviral efficacy of 50% and 90% (Table 1), as observed for other drugs [51].

## Results

### A multiscale model to understand the interplay between treatment, viral dynamics and transmission risk

The multiscale model combines a within-host model of viral dynamics and a between-host model that estimates the probability of virus transmission after a contact [22, 24]. The model reproduces the patterns of kinetics observed with SARS-CoV-2, with a time to peak viral load that coincides with the incubation duration and is equal to 4 days on average, albeit with a large inter-subject variability (Fig 1A), and a median time to viral clearance of 15 days [23]. In this context, initiating a treatment after symptom onset, hence after peak viral load in general, has only a minimal effect on viral load (Fig 1A). Conversely, initiating a treatment before symptom onset can have a dramatic effect on viral load, reducing peak viral load and the duration of viral shedding. Accordingly, treatment initiated after symptom onset is predicted to have a minimal effect on transmission, while a pre-symptomatic treatment may dramatically reduce the risk of transmission (Fig 2). Using a secondary attack rate (SAR) of 40%, as typically reported for SARS-CoV-2 pre-Omicron variants [31], we predict that treatment initiation within 5 days of symptom onset will reduce the mean secondary attack rate to 30%. Treating before symptom onset will reduce SAR to 4.5% (Fig 2C). Similar results are observed for other ranges of SAR, confirming the large benefit of early treatment on reducing virus transmission (Fig 2D).

### Treating index case and household members reduces transmission and virological burden

The model integrates transmission chains to simulate outbreaks in households. In households of size 2 or 3, the Final Attack Rate (FAR), defined as the mean proportion of secondary cases in household (see Methods), and the virological burden, defined as the mean cumulative viral load in household (see Methods), increase linearly with SAR, consistent with the fact that most transmission event originate from the index case. However, in households of size greater than 4, the FAR increases in a non-linear fashion with SAR (Fig 3B), reflecting the multiple chains of transmission that can exist in large households, in particular when SAR has intermediate ranges between 10 and 40%.

Next, we evaluate the impact of antiviral treatment on household transmission. Because antiviral drugs are most generally prescribed with the aim of treating an infected individual, we here consider the most common case where the index case is already symptomatic and is treated within 5 days of symptom onset ($I_{cur}$, see Methods). Because most index cases are already in the clearance phase of the virus when treatment is initiated, this strategy has only a minimal effect in reducing both the number of infected individuals and the virological burden, regardless of SAR (Fig 4) and household size (S2 Fig).

A more aggressive strategy where all household members are treated upon diagnosis of the symptomatic index, regardless of their infection status ($I_{cur} + H_{pep}$, see Methods), is more effective in this case. Although most transmission events originating from the index case have already occurred when treatment is initiated, this strategy acts as a post-exposure prophylaxis, reducing both the risk of further transmission events and the virological burden in secondary infected individuals. Consequently, it leads to a larger reduction in the number of infections,

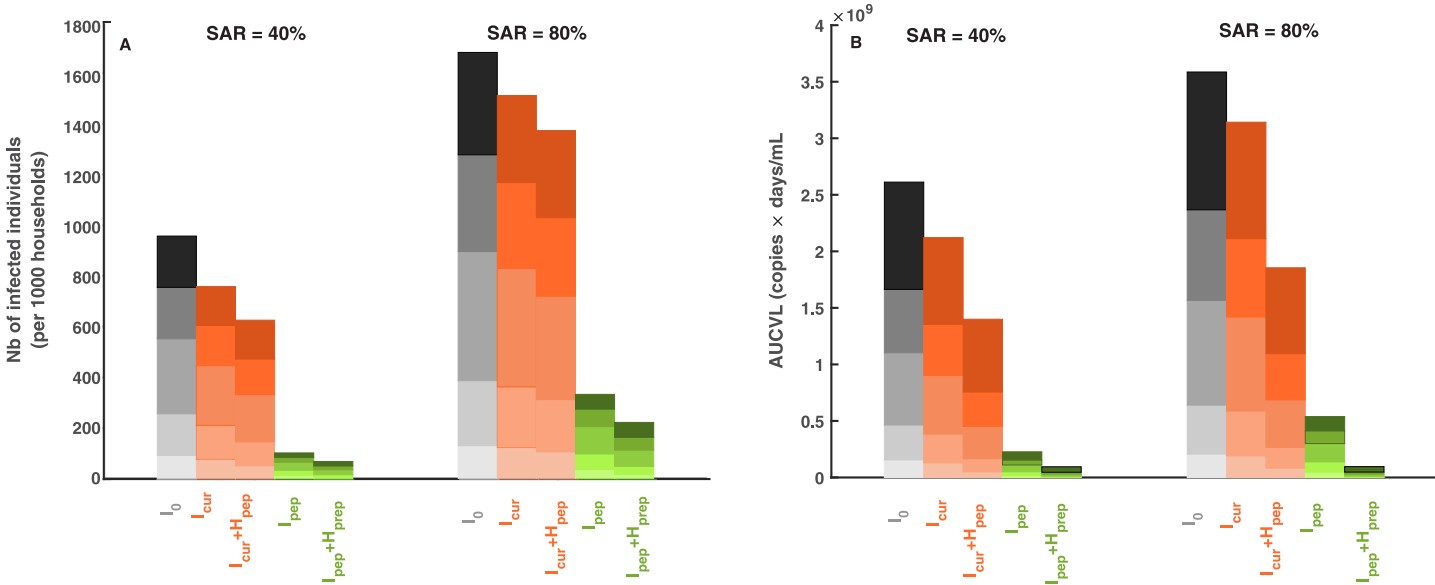

**Fig 4. Impact of treatment strategies on transmission and virological burden for SAR of 40% (left) and 80% (right).** (**A**) Number of infected individuals (per 1000 households); (**B**) Virological burden (per household). Antiviral strategies are the following: no treatment ($I_0$, black), treatment initiated when the index case is symptomatic ($I_{cur}$ and $I_{cur} + H_{pep}$, orange), treatment initiated when the index case is pre-symptomatic ($I_{pep}$ and $I_{pep} + H_{prep}$, green). Results are shown by household sizes, S = 2 (darkest color, top) to S = 6 (lightest color, bottom), with the same distribution of household size than observed in the French population.

in particular in household of size greater than 4, where several transmission chains are more likely to occur (see S2C and S2D Fig). Because only about 25% of households are of size $\geq 4$ in France, the overall effectiveness of this strategy on the number of infections is nonetheless small and remains equal to 20–38% for most cases considered (Fig 5A).

Despite its limited effectiveness on virus transmission, treating all household members can be effective to reduce the virological burden. When baseline SAR is high, e.g. $SAR > 30\%$, the effectiveness ranges 45–52% (Fig 5B). This average effectiveness masks important differences across households composition, with larger effectiveness in households of size $\geq 4$ (see S3A and S3B Fig).

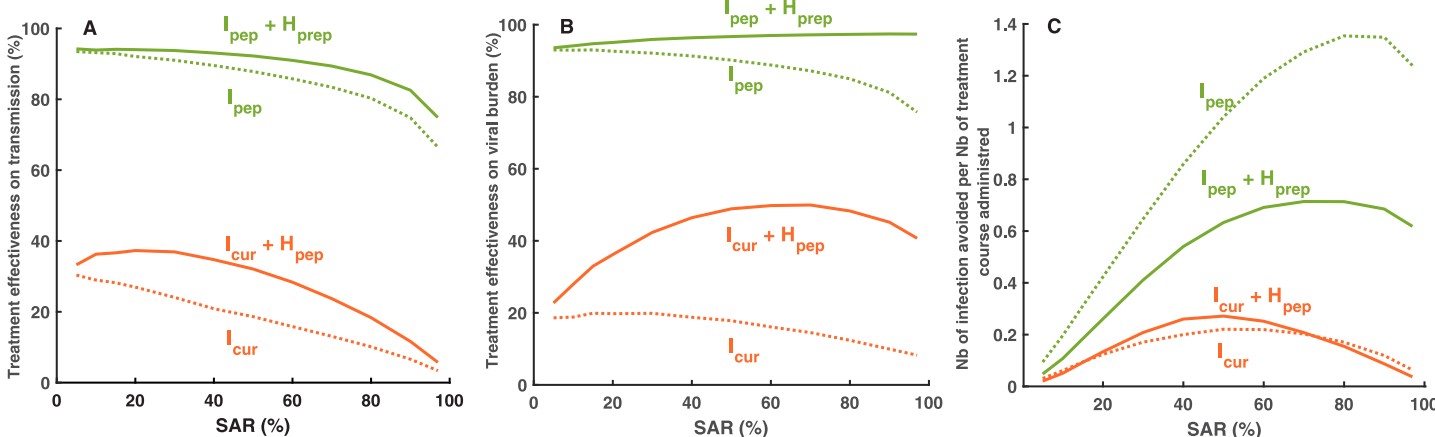

**Fig 5. Effectiveness of treatment strategies.** (**A**) Treatment effectiveness on transmission. (**B**) Treatment effectiveness on virological burden. (**C**) Number of cases avoided per amount of treatments deployed. All effectiveness are relative to no treatment.

### Preempting symptom development in community contacts largely improves the effectiveness of antiviral treatment strategies

We then investigate a strategy in which the index case is treated before symptom onset as post-exposure prophylaxis ($I_{pep}$). Because treatment is initiated before peak viral load, most transmission events are averted. Consequently, this strategy has an effectiveness greater than 75% for most SAR values, and up to 95%, on both the transmission rate and the virological burden (Fig 5), regardless of household size (S3 Fig). Of note the effectiveness of this strategy marks a net decrease for SAR $\geq$ 80%, as the generation time decreases with SAR (Fig 2E and 2F), thereby increasing the risk of virus transmission before treatment initiation (Fig 5A and 5B). Although the additional benefit of treating all household members on virus is low in general (Fig 5), this strategy is relevant to reduce the virological burden, with effectiveness close to 100%, regardless of SAR and household size (Fig 5A and 5B and S3 Fig).

In a cost-effectiveness perspective, treating only the index case before symptom onset is highly efficient (Fig 5C), with a number of cases avoided per treatment unit deployed larger than 1 if SAR is greater than 45%, as compared to a maximum value of 0.27 when the index case is treated after symptom onset (i.e., 4 individuals are treated to avoid 1 infection).

### The model can be directly used to support the design of clinical trials

The modeling framework can also be used to design clinical trials assessing the effectiveness of antiviral strategies (see Methods and S10 and S11 Figs). Focusing on SAR ranging between 20 and 80%, treating only the index case after symptom onset ($I_{cur}$) would require 170–430 households per arm to evidence a clinical effectiveness on transmission (see Fig 6), and 340–450 households to demonstrate effectiveness on virological burden. Treating all household members ($I_{cur} + H_{pep}$) would be more advantageous, requiring 70–240 and 30–100 households to demonstrate a reduction in the transmission rate and the virological burden, respectively.

Much lower numbers are required when the index case is treated in the pre-symptomatic phase ($I_{pep}$ or $I_{pep} + H_{prep}$), with less than 25 households needed to demonstrate an effect on

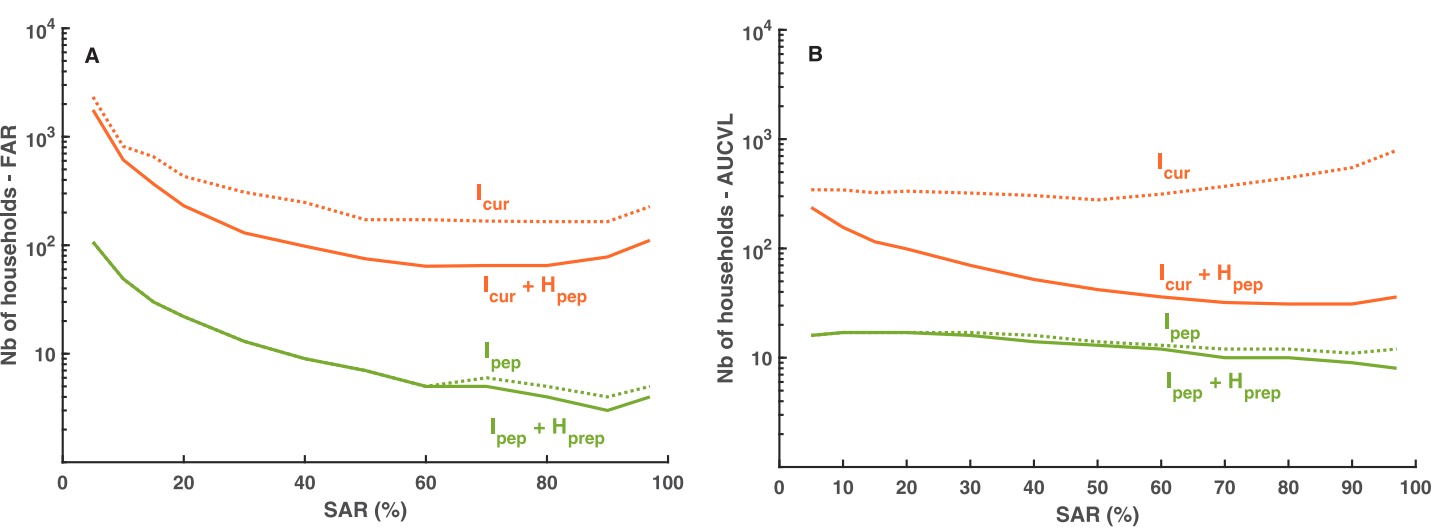

**Fig 6. Number of households (per arm) required to achieve 90% statistical power in 1:1 randomized clinical trial vs placebo (no treatment).** (**A**) Number of households required to demonstrate a reduction in the Final Attack Rate (FAR). (**B**) Number of households (per arm) required to demonstrate a reduction in the virological burden (AUCVL).

the number of transmissions, and less than 20 households to show an effectiveness on the virological burden.

Note that these results are obtained in a context where only 25% of households have more than 4 members (French distribution). Focusing only on large households (see S11B Fig) and treating all their members would diminish the number of households required by a factor of 1.3 and 2.3 for $I_{pep} + H_{prep}$ and $I_{cur} + H_{pep}$ strategies, respectively (S9 Fig).

## Results are robust to changes in SAR expressions and lower levels of antiviral activity

As a sensitivity analysis we also considered the case where SAR decreases with household size (see Methods and S1, S4 and S8 Figs). The results obtained with this alternative model were largely similar to those presented above, which is due to the fact that less than 25% of households are of size 4 or more, thereby limiting the impact of large households (see S4 Fig).

We also evaluated the impact of a treatment with a lower mean antiviral efficacy, noted $\epsilon$, than what was assumed in our baseline scenario ($\epsilon = 99\%$). $\epsilon = 50\%$ leads to a marked decrease of the effectiveness of all strategies (S5 Fig), while $\epsilon = 90\%$ shows almost similar effectiveness in $Icur$ strategies than our baseline scenario. However $\epsilon = 90\%$ leads to a markedly lower effectiveness of $I_{pep}$ strategies (S5 Fig), reducing the clinical effectiveness from more than 75% in the baseline scenario (see above) to between 40 and 80% for most SAR values.

## Treatment effectiveness for other patterns of viral kinetics

Finally, we generalize our approach to different viral kinetics profiles, with a time to peak viral load occurring at 1, 4 and 7 days post infection (S7 Fig), respectively, and a incubation period that remains equal to 4 days [26]. This allows us to generalize our results to different acute viral infections, where the peak viral load can either precede or coincide with symptom onset, as observed for SARS-CoV-2 or RSV [5], or afterwards, as observed for Influenza and SARS-CoV-1 [34–36].

When the index case is already symptomatic ($I_{cur}$ strategies) and peak viral load occurs before or at symptom onset, the effectiveness on reducing transmission remains low in all cases considered (Fig 7). The effectiveness of $I_{cur}$ strategy increases with the time to peak viral load, with values ranging 0–25% when the peak viral load occurs 3 days before symptom onset and 25–75% when the peak viral load occurs 3 days after symptom onset. In all these scenarios, the effectiveness decreases with SAR and we observe a limited additional benefit of treating all household members ($I_{cur} + H_{pep}$). As previously reported in the main scenario, the effectiveness on reducing virological burden is systematically larger than on transmission, in particular when all household members are treated $I_{cur} + H_{pep}$, with effectiveness values ranging from 25–50% when peak viral coincides with symptom onset for SAR < 60%, to values up to 75% when peak viral load occurs more than 3 days after symptom onset.

The effectiveness of strategies reaching out the index case before symptom onset is undoubtedly better ($I_{pep}$ strategies), with effectiveness greater than 75% in all scenarios considered, regardless of SAR, as long as peak viral load coincides or occurs after symptom onset. Notably, the effectiveness is very high even if only the index case is treated. This strategy can even be effective in unfavorable cases where peak viral load precedes symptom onset by three days, with effectiveness in the range 50–75% as long as SAR is less than 70%. Of note these results remain almost equivalent when using the alternative model of SAR according to household size (S8 Fig).

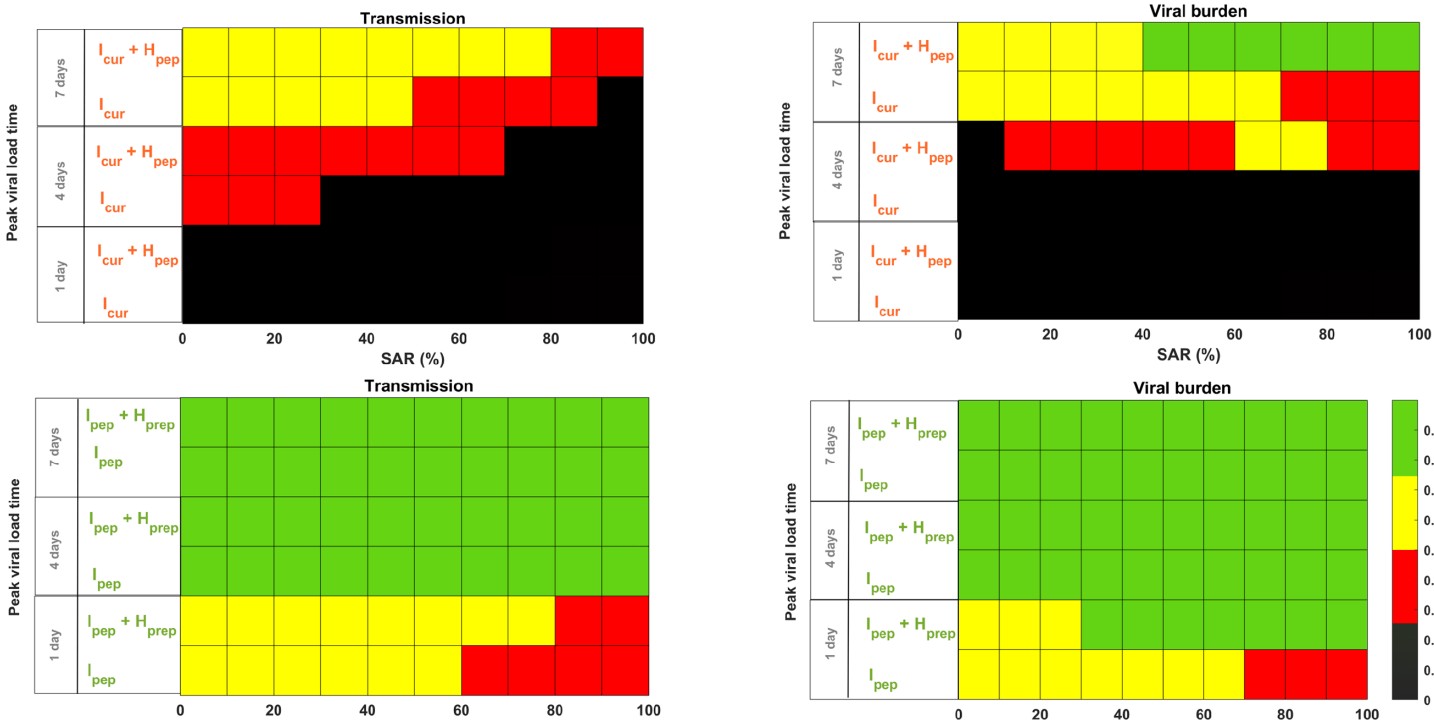

**Fig 7. Effectiveness of treatment strategies according to time to peak viral load.** Left: effectiveness on transmission; right: effectiveness on the virological burden. Top: index case treated after symptom onset ($I_{cur}$ and $I_{cur} + H_{pep}$); bottom: index case treated before symptom onset ($I_{pep}$ and $I_{pep} + H_{prep}$). All effectiveness are relative to no treatment.

## Discussion

Evaluating the global benefit of an antiviral treatment requires not only the quantification of its benefit at the individual level, using virological or clinical endpoints, but also at the population level, in reducing the risk of virus transmission. In the case of acute respiratory viruses, this evaluation is challenging because the risk of transmission is not constant and changes very rapidly over a short period of time, reflecting the dynamic evolution of viral replication. Here, we developed a multi-scale modeling approach that follows viral dynamics at the individual level and relates it to the risk of transmission. Using this framework and applying it to household setting, we could predict the effectiveness of antiviral strategies according to key factors, in particular the virus contagiousness level or what we called the Secondary Attack Rate (SAR), household size, incubation period, time to peak viral load, antiviral efficacy and timing of treatment initiation.

Our results show that in the typical conditions of SARS-CoV-2 pre-Omicron and Omicron variants, with a peak viral load that coincides on average with symptom onset, treatment strategies targeting a symptomatic index case will be poorly effective at reducing both the number of secondary infections and the virological burden. Of note, this result is obtained at the population level, where most households are of size 2 or 3, and masks the role of treatment in large households, with four or more individuals where treating all household members is effective to break transmission chains and reduce virological burden (S3 Fig). In general our results show that when the index case is symptomatic, there is a benefit in treating all household members, regardless of their infection status. This strategy can achieve effectiveness greater than 50% when SAR is less than 80% and peak viral load occurs after symptom onset, as possibly

observed for Influenza [34]. The effectiveness of strategies targeting individuals when they are exposed contact cases, i.e. the index case is treated before symptom onset, is unambiguously better, with effectiveness mostly larger than 50% in all cases considered. In that framework, the additional benefit of treating all household members (of the exposed contact case) is limited, even though a virtually complete abrogation of the virological burden can theoretically be achieved, regardless of SAR. Identifying and treating index cases before symptom onset is challenging, but is not out of reach. During the last pandemic, most clinical trials focused on outpatients within 5 or 7 days of symptom onset [12], or on pre-exposure prophylaxis, but some studies enrolled high risk contact patients [11, 44]. In future outbreaks, the experience acquired on contact tracing, rapid drug distribution, and implementation of contactless clinical studies [42] will improve the capability to run reactive clinical trials in the community, provided that affordable, safe and oral drugs are available. In that perspective, our results show that strategies targeting the index case are key to achieve a high effectiveness, and could participate to the arsenal of public health measures to reduce the burden of disease, protect most at risk individuals and accordingly reduce the duration of isolation of high risk contacts.

This modeling framework can also be used to design clinical trials. As an illustration, we calculated the number of households required to assess effectiveness on either transmission or virological burden, assuming a 1:1 household randomized study between treatment and placebo. Because of their relatively low effectiveness, strategies enrolling only symptomatic index would require about 170–430 households (per arm) to assess effectiveness on transmission. Although these numbers may appear to be high, they remain doable, in particular if considering contactless trial [42]. More sophisticated designs could also be proposed, where the randomization could be at the individual rather than at the household level, or where different treatment strategies could be proposed in index and contact cases. To keep our results easily interpretable we did not consider this possibility here. Such policy would require to address ethical and practical challenges, in particular the risk of pill sharing. Also, our results show that study endpoints focusing on virological burden are systematically more powerful than those focusing on transmission, reducing the number of needed household to less than 100 for SAR greater than 20%. However, a virological endpoint has also practical drawbacks, in particular the requirement to draw repeated PCR tests in all household members. Note that, here as well, focusing on households of size of 4 and larger may be more powerful.

Our model makes a number of important assumptions and simplifications that we discuss below.

First, while our central scenario relied on typical parameters of SARS-CoV-2 viral dynamics, our approach aims to provide a general understanding of the factors governing treatment efficacy against ARI and is not tailored for one specific virus. Indeed, we explored a variety of viruses, as illustrated by the large range of SAR values covering different possible ranges of virus transmissibility, and the different times to peak viral load, close to other ARI such as Influenza or RSV. Although vaccination or new variants can affect the within-host viral dynamics, this question was not in the scope of this study but the model can easily be adapted to any host-pathogen characteristics by changing within-host parameters. Further, a power law model was used to relate the viral load at time $t$ and the risk of transmission (see Eq (2)). Although this model was successfully used to reproduce the risk of transmission in SARS-CoV-2 [24], and that we here explored a large range of parameter values to encompass different levels of virus transmissibility, this model may not universally apply to all ARI. Such differences could for instance reflect the importance of the types of droplets causing the infection and/or different symptom patterns leading to different functional forms between viral load and transmission.

Our model assumes that SAR is independent of household size. Although the results of epidemiological studies are heterogeneous, they generally report that SAR decreases with household size [19, 27–30], possibly reflecting the increased heterogeneity of contacts between members of different ages in large households. We therefore investigated an alternative model in which larger households leads to lower SAR. This assumption did not affect highly our results (see S8 Fig), as less than a third of our households has four or more members in our population. More generally we assumed homogeneous mixing the household, and did not consider more realistic contact patterns. In the future, model refinement relying directly on contact data [37] could be used to modulate the risk of virus acquisition according to individual demographic factors, such as age, type of relationship (siblings, parents) [38–41], or viral dynamic factors [21]. Taking into account individual heterogeneity would also be relevant to evaluate the benefit of antiviral treatments on other endpoints, such as the risk of severe disease, and to adjust for pre-existing immunity and vaccine intake. Another important assumption of the model is that infectiousness is directly related to viral load levels in the nasopharynx, albeit in a non-linear fashion and with large inter-individual variability. Although we and others have shown that higher viral load at the time of a high risk contact is associated with a higher risk of transmission [24, 25], it should be recognized that a formal quantification of the impact of nasopharyngeal viral load on the risk of transmission is still lacking. Likewise, fomites, or simply viral dynamics in other compartments, such as saliva, could be relevant when it comes to household transmission, even though its relationship with infectiousness has not been established [43]. Second, our results emphasize the benefit of strategies focusing on pre-symptomatic index case ($I_{pep}$). Treating invididuals in post exposure prophylaxis however increases the risk to treat uninfected individuals, and in turn increases the risk of over-treatment. While we recognize that this may lead to overestimate the cost-effectiveness of $I_{pep}$ strategies, we note that this could be mitigated by including only individuals with a positive PCR test, as the time between infection and first positive nasal swab PCR test is very short and is between 1 and 3 days for Influenza or SARS-CoV-2 viruses [34, 36, 46].

To conclude our model can be used to quantify and anticipate the clinical effectiveness of antiviral treatment strategies against acute respiratory viruses in households. It provides a novel understanding on the conditions that need to be met, at the pharmacological, virological and behavioural levels, for an antiviral treatment to be effective, and can guide interventions aiming to reduce disease burden during a viral pandemics.

## Supporting information

**S1 Text. Viral load dynamic model and calibration.**
(PDF)

**S1 Fig. Impact of modified SAR, $SAR_S$, on the attack rate.** Left figure is the FAR according to SAR in the modified SAR model for the different household sizes (darkest gray for S = 2 to lightest S = 6) and averaged over France household statistics with the non-modified SAR model (yellow) and with the modified one (red). The right figure represents Mean estimate of the household reproductive number, $R_0$, (computed over the French distribution of household sizes) as a function of SAR. $R_0$ represents the mean number of individuals infected directly by the index case in a household. The red line represents the value obtained using the modified $SAR_S$ model; the yellow line represents the value obtained using the non-modified SAR model. (EPS)

**S2 Fig. Transmission and virological burden according to household size for different treatment timings.** (A,C,E,G) Number of infected individuals per 1000 households. (**B,D,F,**

**H**) Virological burden. Gray: $I_0$; orange dashed line: $I_{cur}$; orange line: $I_{cur} + H_{pep}$; green dashed line: $I_{pep}$; green line: $I_{pep} + H_{prep}$. Results are shown by household sizes, S = 2 (darkest color) to S = 6 (lightest color).
(PDF)

**S3 Fig. Treatment Effectiveness according to household size.** (**A,D**) Treatment effectiveness on transmission. (**B,E**) Treatment effectiveness on virological burden. (**C,F**) Number of cases avoided per amount of treatments deployed. Gray: $I_0$; orange dashed line: $I_{cur}$; orange continuous line: $I_{cur} + H_{pep}$; green dashed line: $I_{pep}$; green continuous line: $I_{pep} + H_{prep}$. Results are shown by household sizes, S = 2 (darkest color) to S = 6 (lightest color).
(EPS)

**S4 Fig. Treatment Effectiveness according to the different scenarios for the alternative SAR definition $SAR_S$.** (**A**) Treatment effectiveness on transmission. (**B**) Treatment effectiveness on virological burden.
(EPS)

**S5 Fig. Viral dynamics for different antiviral efficacies.** Top: $\epsilon$ = 50%; bottom: $\epsilon$ = 90%. (**A, D**) Individual viral dynamic profiles predicted by the model in 30 individuals (see Eq (3)), that are either left untreated ($I_0$, gray), treated within 5 days after symptom onset ($I_{cur}$, orange), or treated symptom onset ($I_{pep}$, green). All parameters are given in S1 Table. (**B,E**) Distribution of the peak viral load predicted by the model. (**C,F**) Distribution of the time to peak viral load predicted by the model.
(EPS)

**S6 Fig. Effectiveness of treatment strategies depending on the antiviral efficacy.** (**A,D**) Treatment effectiveness on transmission. (**B,E**) Treatment effectiveness on virological burden. (**C,F**) Number of cases avoided per amount of treatments deployed. Plain line: $\epsilon$ = 99%; Diamond line: $\epsilon$ = 90%; Square line: $\epsilon$ = 50%; Orange dashed line: $I_{cur}$; Orange continuous line: $I_{cur} + H_{pep}$; Green dashed line: $I_{pep}$; Green continuous line: $I_{pep} + H_{prep}$. All effectiveness are relative to no treatment.
(EPS)

**S7 Fig. Viral dynamics and antiviral treatment for mean time to viral load peak 1d and 7d.** Individual viral dynamic profiles predicted by the model in 30 individuals (see Eq (1) and S3 Table), that are either left untreated $I_0$ (gray), treated within 5 days after symptom onset $I_{cur}$ (orange), or treated before symptoms onset $I_{pep}$ (green). All parameters are given in S1 and S3 Tables, and the model assumes that a mean incubation period of 4 days, and a mean treatment antiviral efficacy of 99%. Top: Mean time to viral load peak = 1d. Bottom: Mean time to viral load peak = 7d.
(EPS)

**S8 Fig. Effectiveness of treatment strategies according to time to peak viral load with modified $SAR_S$ according to household sizes S.** Left: effectiveness on transmission; right: effectiveness on the virological burden. Top: index case treated after symptom onset ($I_{cur}$ and $I_{cur} + H_{pep}$); bottom: index case treated before symptom onset ($I_{pep}$ and $I_{pep} + H_{prep}$). All effectiveness are relative to no treatment.
(EPS)

**S9 Fig. Number of households of size 4 and larger required to achieve 90% statistical power in 1:1 randomized clinical trial vs placebo (no treatment).** (**A**) Number of households required to demonstrate a reduction in the Final Attack Rate (FAR). (**B**) Number of

households required to demonstrate a reduction in the virological burden (AUCVL).
(EPS)

**S10 Fig. Distribution of transmission rate ($FAR_h$) and virological burden ($AUCVL_h$) for SAR = 40% and different household sizes $S$.** Gray: $I_0$; Orange: $I_{cur} + H_{pep}$; Green: $I_{pep} + H_{prep}$.
(EPS)

**S11 Fig. Mean value and standard deviation for $AUCVL_h$.** These values were calculated for all household size statistics (A, upper figures), and for households of size 4 and larger (B, bottom figures). Gray: untreated households, and for treated households, Orange dashed line: $I_{cur}$; Orange continuous line: $I_{cur} + H_{pep}$; Green dashed line: $I_{pep}$; Green continuous line: $I_{pep} + H_{prep}$.
(EPS)

**S1 Table. Parameter values used to generate viral dynamic profiles and transmission probabilities.**
(PDF)

**S2 Table. Proportion of household sizes in France [33].**
(PDF)

**S3 Table. Modified parameters in the viral load model (see Eq (1)) to reproduce time to peak viral load equal to 1 and 7 days.**
(PDF)

**S1 Code. Matlab code and code description in *README file.pdf*.** The code is protected by Agence de Protection des Programmes (APP) in France, and we have the permission to share it.
(ZIP)

## Acknowledgments

The authors thank France Mentré, Evelina Tacconelli and Fabrice Carrat for providing helpful comments on a previous version of the manuscript.

## Author Contributions

**Conceptualization:** Hind Zaaraoui, Lulla Opatowski, Jérémie Guedj.

**Data curation:** Clarisse Schumer.

**Funding acquisition:** Jérémie Guedj.

**Investigation:** Hind Zaaraoui, Xavier Duval, Bruno Hoen, Lulla Opatowski, Jérémie Guedj.

**Methodology:** Hind Zaaraoui, Clarisse Schumer, Lulla Opatowski, Jérémie Guedj.

**Project administration:** Jérémie Guedj.

**Visualization:** Hind Zaaraoui.

**Writing – original draft:** Hind Zaaraoui, Xavier Duval, Bruno Hoen, Lulla Opatowski, Jérémie Guedj.

**Writing – review & editing:** Hind Zaaraoui, Xavier Duval, Bruno Hoen, Lulla Opatowski, Jérémie Guedj.

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
