## [Decision Letter · Decision Letter 0]

9 Nov 2023

Dear Zaaraoui,

Thank you very much for submitting your manuscript "Modelling the effectiveness of antiviral treatment strategies to  prevent household transmission of acute respiratory viruses" for consideration at PLOS Computational Biology.

As with all papers reviewed by the journal, your manuscript was reviewed by members of the editorial board and by several independent reviewers. In light of the reviews (below this email), we would like to invite the resubmission of a significantly-revised version that takes into account the reviewers' comments.

The Authors are expected to address all the criticisms by all Reviewers. In particular, perform validation of the key parameters, assumptions and model output (Reviewer #1), clarify which parameters were estimated, confirm and clarify the viral load data from reference 20, clarify how the timing of treatment initiation was defined and state clearly the definition of effectiveness (Reviewer #2). In additional to the above comments, please address,

1. Table S1, please provide the references for all parameters which were not estimated from data

2. The main study finding is sensitive to viral dynamics and epidemiological characteristics of SARS-CoV-2, which has been changing between variants. For example, the peak viral load for omicron variant appeared several days after symptom onset (PMID: 37768707). For relevance, the authors should consider using parameters from more recent variants.

We cannot make any decision about publication until we have seen the revised manuscript and your response to the reviewers' comments. Your revised manuscript is also likely to be sent to reviewers for further evaluation.

Sincerely,

Eric HY Lau, Ph.D.

Academic Editor

PLOS Computational Biology

Amber Smith

Section Editor

PLOS Computational Biology

The Authors are expected to address all the criticisms by all Reviewers. In particular, perform validation of the key parameters, assumptions and model output (Reviewer #1), clarify which parameters were estimated, confirm and clarify the viral load data from reference 20, clarify how the timing of treatment initiation was defined and state clearly the definition of effectiveness (Reviewer #2). In additional to the above comments, please address,

1. Table S1, please provide the references for all parameters which were not estimated from data

2. The main study finding is sensitive to viral dynamics and epidemiological characteristics of SARS-CoV-2, which has been changing between variants. For example, the peak viral load for omicron variant appeared several days after symptom onset (PMID: 37768707). For relevance, the authors should consider using parameters from more recent variants.

Reviewer's Responses to Questions

**Comments to the Authors:**

Reviewer #1: This study aimed to evaluate the impact of antiviral treatment on the transmission of infectious diseases within households. The authors discovered that initiating treatment before the peak viral load, which coincides with the onset of symptoms of ancestral SARS-CoV-2, resulted in a high level of effectiveness for the intervention. Therefore, targeting households with pre-symptomatic individuals could potentially reduce the burden of the disease. Additionally, the proposed theoretical and computational framework shows promise in assessing the effectiveness of antiviral treatment for various infectious diseases transmitted within households.

While the study is theoretically interesting and holds potential for practical application in public health and clinical science, it heavily relies on viral load data alone. To strengthen its validity, I recommend validating the proposed computational approach using household epidemiological or clinical data. Although I recognize the value of this study as a simulation work, interpreting its findings for public health practice may present challenges or even pose risks. Regrettably, I am unable to endorse the publication of this paper in Nature Communications.

[Major comments]

・The proposed multiscale model consists of a viral dynamics model and a transmission model, each parameterized using different datasets or assumptions. As a result, I find that both the model structure and the values assigned to its parameters lack complete validation.

・The model incorporates numerous assumptions, making it difficult for readers to discern which assumptions are reasonable or less reasonable, and how they may influence the study's conclusions.

Reviewer #2: The authors attempted a comprehensive investigation of the prophylactic efficacy of oral antivirals by simulating the within-host virus dynamics, estimated from the viral load of over 4000 patients, combined with the probability of transmission in the household, estimated from data in previous studies. They provided a computational framework for evaluating the effect of prophylactic administration of antiviral drugs in French family structures in terms of virus transmission and cumulative viral load by numerical simulations considering individual differences in virus transmission dynamics and contact heterogeneity. It evaluated the differences in effectiveness of treatment strategies determined by a combination of the timing of administration based on before and after the onset of index cases (in this case, corresponding to before and after the peak of viral load) and the selection of the target of administration, and also provided important insights into the variation in sample size when conducting clinical trials. In particular, the study shows how the effect varies with the Secondary Attack Rate, which can be assessed epidemiologically, and might be important from a public health perspective for considering countermeasures against emerging infectious diseases. While the overall assertions of the paper are relatively clear, there appears to be a lack of description of the methodology and original data from which their results were derived, and I recommend that the manuscript be revised to clarify the basis and accuracy of the assertions. My concerns of this article are as follows.

Major comments:

1. In the model with antiviral treatment placed between Lines 175 and 176, the description of I_t>tx is not provided. The definition of this variable may change the meaning of the model, so please add it to the text. And tx must be subscripted as t_x.

2. Regarding the population parameters in the S1 Table, information on which parameters were estimated from the data is missing from the text and needs to be provided. It also appears that there are no published values for the patient's viral load from the reference 20 (Conrado JD et al. 2022). If so, it seems that the data availability needs to be corrected or data should be provided according to journal guidelines. At least, the authors need to show what the data fitting results in.

3. No information is provided on how to choose a value for t_x, the time to start treatment. Just the treatment strategy is specified as treatment before onset and within 5 days of symptom onset, however, how t_x is varied in this context is important information. Relatedly, the choice of incubation period tau is unclear: S1 Table says it is log-normally distributed, but looking at S6 Fig, it appears that there are no cases with incubation period shorter than 3 days, as described in the text. Does this mean that the lognormal distribution has been changed to a left-truncated distribution?

4. In Figures 2B-C, P(t) and SAR are calculated for the no treatment and treatment cases (and treatment strategies), respectively, while in the subsequent results the SAR is fixed (i.e., using the value of the SAR for the no treatment case) and the effect of antiviral drugs is discussed. To avoid reader confusion, please eliminate the mention of P(t) and SAR in Figure 2 or separate the definitions of SAR for no treatment and treatment in the Methods section. Alternatively, for results after Figure 2, it should be explicitly stated in all corresponding sections that the results are for the no treatment SAR.

5. Referring to the definition of AUCVL in equation (5), it appears that the variable C is the number of infected cases for each index case (each household). Under this definition, the "FAR" provided in Line 333 and Figure 3B appears to be the expected value of the FAR, so if necessary, please explicitly state "expected value of FAR" in the corresponding location. If this understanding is correct, please also add it to the definition of C on Line 219.

6. Please provide the definition of "effectiveness" in Figure 5 and Figure 7. The interpretation may vary depending on whether the number of infections or AUCVL for no treatment or treatment is calculated and compared within families with the same viral dynamics parameters, or whether the means of the two groups are compared. In particular, I think it is also necessary to include the procedure how the effect of prevention of infection per unit of treatment, described in Lines 388 to 392, is excluded.

7. Please provide the mean and variance of the FAR and AUCVL from which the authors calculated the sample size in Figure 6, as both affect the calculation of sample size using the t-test and thus help interpret the results of the sample size changes obtained. Also, the discussion of clinical trials restricted to families with four or more family members, shown in Lines 412 to 414 and Lines 522 and 525, is not an appropriate discussion if the variances are not provided. More to this point, the sample size in the clinical trial when limited to families with four or more members needs to be shown for this discussion.

8. How is the treatment effectiveness calculated for the modified SAR shown in S4 Fig and S7 Fig? According to the definition of modified SAR given in the Methods section, this value seems to vary depending on the number of people in the household. In other words, even if one parameter m is chosen to define the distribution of M representing the impact of viral load on transmission, the SAR value will change depending on the number of people in the household. In the unmodified SAR, the SAR is uniquely determined by fixing the parameter m, against which the treatment effectiveness is calculated by the distribution of the number of people in French households. However, under a SAR that varies with the number of people in the household, it appears that treatment effectiveness cannot be calculated under a particular modified value of SAR.

9. Please provide viral dynamics as shown in Figure 1 and S6 Fig for the discussion of the impact of multiple treatment effects, epsilon = 0.9 or 0.5 in lines 424 to 434.

10. In Lines 435 – 563 and Figure 7, the delay between symptom onset and time to peak viral load is not appropriate to explain the differences in viral infection dynamics because both the incubation period and time to peak may affect on effectiveness. For the sake of consistency in the discussion, I recommend showing this as the time of the peak of the viral load, as shown in Lines 294-304 and S3 Table.

Minor comments:

1. What dose R_0 mean in S1. Fig? It seems to be different from R_0 in S1 Table, which is calculated from model parameters and not depending on SAR.

2. The simultaneous equations between lines 175 and 176 are not numbered in the equation, but the reference elsewhere is to Eq. (5), which should be corrected.

3. In Line 182 and 184 and S1 Table, epsilon is shown as a percentage value, but in the mathematical model definition, it appears to be between 0 and 1. If this is correct, I recommend that it be stated accurately, i.e. epsilon = 0.99, 0.9 or 0.5.

4. In Line 192, please revise “see (2)” to “see Eq. (2)”.

5. In Line 196, please delete the duplicate "that".

6. Please add an explanation of the meaning of “grey” for Figure 1BC. The presence of the gray distribution may be missed because it overlaps with other colors.

7. It would be helpful to refer to the panel of figures to aid the reader's understanding. In Line 363, “S3 Fig” can be “S3A and B Fig”, in Line 373, 380 “S3 Fig” can be “S3D and E Fig”, and in Line 407, “Fig. 6” can be “Fig. 6B”. Please check the journal's guidelines for reference style of figures.

8. Please specify that beta_5d, delta_5d, p_5d in S3 Table is the same as beta, delta, p in S1 Table.

9. In my environment, there was a garbled character in line 386. Looks “S¿4”, this might be “S > 4”.

10. In Lines 405-407, please clarify that this sentence is for treatment initiated after symptom of index.

11. In Line 437, referring Figure 1 for cases with peak viral load at 5 days.

12. In S6 Fig please provide the difference of colors as provided in Figure 1

13. In Lines 445 – 448, Lines 450 – 454, are the percentage values (25-50% and 50-75%) shown here SAR?

**Have the authors made all data and (if applicable) computational code underlying the findings in their manuscript fully available?**

Reviewer #1: **No: **

Reviewer #2: **No: **The data from which model parameters were estimated are not in reference or manuscript, or explanation seems lacking.

PLOS authors have the option to publish the peer review history of their article (what does this mean?). If published, this will include your full peer review and any attached files.

Reviewer #1: No

Reviewer #2: **Yes: **Shoya Iwanami
---

## [Decision Letter · Decision Letter 1]

10 Apr 2024

Dear Zaaraoui,

Thank you very much for submitting your manuscript "Modelling the effectiveness of antiviral treatment strategies to  prevent household transmission of acute respiratory viruses" for consideration at PLOS Computational Biology. As with all papers reviewed by the journal, your manuscript was reviewed by members of the editorial board and by several independent reviewers. The reviewers appreciated the attention to an important topic. Based on the reviews, we are likely to accept this manuscript for publication, providing that you modify the manuscript according to the review recommendations.

The Authors have improved the analyses, provided supporting results concerning the validity of the connection between viral dynamics and transmission, and clarified the model assumptions. The Authors are expected to address all further criticisms by Reviewer #1. In particular, please clarify the start time of treatment and C_[0,t].

Sincerely,

Eric HY Lau, Ph.D.

Academic Editor

PLOS Computational Biology

Amber Smith

Section Editor

PLOS Computational Biology

The Authors have improved the analyses, provided supporting results concerning the validity of the connection between viral dynamics and transmission, and clarified the model assumptions. The Authors are expected to address all further criticisms by Reviewer #1. In particular, please clarify the start time of treatment and C_[0,t].

Reviewer's Responses to Questions

**Comments to the Authors:**

Reviewer #2: Thank you for your sincere response to my comments and understanding of the differences. Most of my concerns have been addressed, but I would appreciate it if you would consider correcting the remaining minor points.

1. Terminology

In the 3rd to 6th lines in Abstract, “During an outbreak, more aggressive 16 strategies involving pre- or post-exposure prophylaxis (PEP or PrEP) could be 17 employed to further reduce the risk of severe disease but also prevent transmission to 18 household contacts.”, "PEP" and "PrEP" may be opposite.

2. Generality of the model

I think it is better to mention as a limitation that the transmission model (p(t)) formula itself is not a complete generalization of respiratory infectious diseases, since it is estimated from COVID-19 data and has not been validated in other viral infections. Even among respiratory infections, the importance of the types of droplets and their symptoms may differ among different types of infections.

3. Details of the Methods

P(∞) of the SAR should be calculated at a time large enough to be approximated as P(t_∞) in practice, so please add it to the Transmission model in Methods.

It seems that C_[0,t], which is required for the calculation of SAR, is not provided.

I could not find detailed information on how the timing of individual treatments is chosen for t_x, the start time of treatment. Table 1 only provides information on whether it is before or after symptom onset. t_x may be a constant, or it may be randomly sampled from some distribution, Either way, the authors need to specify how the actual values used were determined. This needs to be provided for the simulations in Fig. 1 and Fig. 2 and for the case of household infection after Fig. 4, if different from each other.

4. The sample size calculation

Fig. S10 may be the distribution of FAR_h and AUCVL_h.

The sample size discussion does not seem to be significantly affected by the variance of AUCVL_h as shown in the Letter. However, the sample size change in relation to SAR is not only determined by the AUCVL change, so I think it should be included in the supplementary information. I also think that the values of AUCVL and the standard deviation of AUCVL_h should be shown for the case restricted to households with 4 or more members, which you added to S9 Fig.

**Have the authors made all data and (if applicable) computational code underlying the findings in their manuscript fully available?**

Reviewer #2: **No: **There is no published data in the reference on the amount of viruses that can be used as a basis for the study. However, this point may be resolved at a later date. All information necessary for the calculations is disclosed in the text.

PLOS authors have the option to publish the peer review history of their article (what does this mean?). If published, this will include your full peer review and any attached files.

Reviewer #2: **Yes: **Shoya Iwanami

Figure Files:

Data Requirements:

Reproducibility:

References:

---

## [Decision Letter · Decision Letter 2]

19 Jul 2024

Dear Zaaraoui,

Thank you very much for submitting your manuscript "Modelling the effectiveness of antiviral treatment strategies to  prevent household transmission of acute respiratory viruses" for consideration at PLOS Computational Biology.

As with all papers reviewed by the journal, your manuscript was reviewed by members of the editorial board and by several independent reviewers. In light of the reviews (below this email), we would like to invite the resubmission of a significantly-revised version that takes into account the reviewers' comments.

The authors have addressed most of the reviewers and editor’s comments. However, a common major comment from more than one reviewers concerning the viral kinetics parameters remains unresolved (confidential comments). Specifically, for the parameters the authors cited a reference #20 which was not a preprint, published or accepted manuscript. The study (ref #20) investigated the viral kinetics of COVID patients treated with casirivimab+imdevimab. Please be aware that citing a conference abstract as the source of data that is critical for the model is not acceptable per journal policies as it is important for other researchers to be able to validate/reproduce the work. It appears that the conference presentation described the estimation of parameters based on data from two clinical trials that were published. However, this is not entirely clear. If the data has been published or is otherwise publicly available, please include reference(s) to those original publications. If the data has not been published, the data either needs to be included as original research or the data and any accompanying analysis will need to be removed from the manuscript. If the analysis has since been published, please update the manuscript references accordingly. If the model fitting to this data has not been published elsewhere and only appears in the conference abstract, the methods and parameters (currently reported in Table S1) need to be more clearly detailed here as original research. Please also include your code for the analysis per journal policies. Alternatively, a reference for the parameters could be another antiviral treatment as the analysis was not targeted at a specific treatment.

Another unresolved critique is that the study objective investigates the effectiveness of antiviral treatment strategies to prevent household transmission of acute respiratory viruses, but the analysis and parameterization were mostly built on pre-omicron SARS-CoV-2. The authors should revise the title, abstract and text accordingly, while discussing the generalizability of the framework or results to more general acute respiratory viruses.

We cannot make any decision about publication until we have seen the revised manuscript and your response to the reviewers' comments. Your revised manuscript is also likely to be sent to reviewers for further evaluation.

Sincerely,

Eric HY Lau, Ph.D.

Academic Editor

PLOS Computational Biology

Amber Smith

Section Editor

PLOS Computational Biology

The authors have addressed most of the reviewers and editor’s comments. However, a common major comment from more than one reviewers concerning the viral kinetics parameters remains unresolved (confidential comments). Specifically, for the parameters the authors cited a reference #20 which was not a preprint, published or accepted manuscript. The study (ref #20) investigated the viral kinetics of COVID patients treated with casirivimab+imdevimab. Would the authors be able to either provide an update of the study in a form of preprint or manuscript for reference #20, or provide a reference accepted by the journal (https://journals.plos.org/ploscompbiol/s/submission-guidelines#loc-references)? Please note that the reference for the parameters could be other antiviral treatment as the analysis was not targeted at a specific treatment.

Also, the study objective investigates the effectiveness of antiviral treatment strategies to prevent household transmission of acute respiratory viruses. However, the analysis and parameterization were mostly built on pre-omicron SARS-CoV-2. The authors should revise the title, abstract and text accordingly, while discussing the generalizability of the framework or results to more general acute respiratory viruses.

Reviewer's Responses to Questions

**Comments to the Authors:**

Reviewer #2: All my concerns were addressed except for the original clinical trial data and the process of estimating the parameters of the mathematical model.

Reviewer #3: This manuscript develops a multi-scale model to evaluate the effectiveness of antiviral treatment strategies in reducing the transmission of acute respiratory viruses, such as SARS-CoV-2, within household settings. The study explores various antiviral strategies, including pre- and post-exposure prophylaxis, and assesses their impact on transmission rates and virological burden in different household sizes. The findings suggest that early treatment, especially before symptom onset, significantly reduces both transmission and virological burden, highlighting the potential of antiviral interventions in controlling outbreaks within households.

The integration of individual-level viral dynamics and population-level transmission dynamics is surely a strength of this paper. It includes a thorough simulation study that explores a wide range of virus characteristics and treatment scenarios, providing robust insights into the effectiveness of various strategies.

The findings are well-presented with clear figures and tables, making it easy to understand the impact of different treatment strategies on transmission and virological burden.

Although I did not review the previous versions of this submission, I feel that the authors have done a good job in addressing comments, some of which would also be mine. I recommend its publication.

**Have the authors made all data and (if applicable) computational code underlying the findings in their manuscript fully available?**

Reviewer #2: **No: **There is no published reference on data on viral load in the two clinical trials or on the methods used to estimate the parameters of the mathematical model.

Reviewer #3: Yes

PLOS authors have the option to publish the peer review history of their article (what does this mean?). If published, this will include your full peer review and any attached files.

Reviewer #2: **Yes: **Shoya Iwanami

Reviewer #3: No
---

## [Editor Report · Decision Letter 3]

18 Oct 2024

Dear Zaaraoui,

We are pleased to inform you that your manuscript 'Modelling the effectiveness of antiviral treatment strategies to  prevent household transmission of acute respiratory viruses' has been provisionally accepted for publication in PLOS Computational Biology.

Best regards,

Eric HY Lau, Ph.D.

Academic Editor

PLOS Computational Biology

Amber Smith

Section Editor

PLOS Computational Biology

Thanks for addressing all the editor’s and reviewers' comments. Congratulations on the excellent work!

---

## [Editor Report · Acceptance letter]

14 Nov 2024

PCOMPBIOL-D-23-01419R3 

Modelling the effectiveness of antiviral treatment strategies to  prevent household transmission of acute respiratory viruses

Dear Dr Zaaraoui,

I am pleased to inform you that your manuscript has been formally accepted for publication in PLOS Computational Biology. Your manuscript is now with our production department and you will be notified of the publication date in due course.

With kind regards,

Anita Estes
